# Prediction of Oscillations in Glycolysis in Ethanol-Consuming Erythrocyte-Bioreactors

**DOI:** 10.3390/ijms241210124

**Published:** 2023-06-14

**Authors:** Evgeniy Protasov, Michael Martinov, Elena Sinauridze, Victor Vitvitsky, Fazoil Ataullakhanov

**Affiliations:** 1Laboratory of Biophysics, Dmitriy Rogachev National Medical Research Center of Pediatric Hematology, Oncology, and Immunology, Ministry of Healthcare, Samora Mashel Str., 1, GSP-7, Moscow 117198, Russia; protasov_evgenii@mail.ru (E.P.); ataullakhanov.fazly@gmail.com (F.A.); 2Laboratory of Physiology and Biophysics of the Cell, Center for Theoretical Problems of Physicochemical Pharmacology, Russian Academy of Sciences, Srednyaya Kalitnikovskaya Str., 30, Moscow 109029, Russia; martinov.michael@gmail.com; 3Department of Molecular and Translational Medicine, Moscow Institute of Physics and Technology, Institutskiy Per., 9, Dolgoprudny 141701, Russia; 4Perelman School of Medicine, University of Pennsylvania, 3400 Civic Center Blvd., Philadelphia, PA 19104, USA

**Keywords:** acetaldehyde dehydrogenase, alcohol dehydrogenase, erythrocyte-bioreactor, ethanol, glycolysis, mathematical model, NAD, oscillations

## Abstract

A mathematical model of energy metabolism in erythrocyte-bioreactors loaded with alcohol dehydrogenase and acetaldehyde dehydrogenase was constructed and analyzed. Such erythrocytes can convert ethanol to acetate using intracellular NAD and can therefore be used to treat alcohol intoxication. Analysis of the model revealed that the rate of ethanol consumption by the erythrocyte-bioreactors increases proportionally to the activity of incorporated ethanol-consuming enzymes until their activity reaches a specific threshold level. When the ethanol-consuming enzyme activity exceeds this threshold, the steady state in the model becomes unstable and the model switches to an oscillation mode caused by the competition between glyceraldehyde phosphate dehydrogenase and ethanol-consuming enzymes for NAD. The amplitude and period of metabolite oscillations first increase with the increase in the activity of the encapsulated enzymes. A further increase in these activities leads to a loss of the glycolysis steady state, and a permanent accumulation of glycolytic intermediates. The oscillation mode and the loss of the steady state can lead to the osmotic destruction of erythrocyte-bioreactors due to an accumulation of intracellular metabolites. Our results demonstrate that the interaction of enzymes encapsulated in erythrocyte-bioreactors with erythrocyte metabolism should be taken into account in order to achieve the optimal efficacy of these bioreactors.

## 1. Introduction

Erythrocytes with encapsulated enzymes that are not present naturally in these cells are known as erythrocyte-bioreactors (EBRs) and may be used for many biological and medical tasks [1,2]. Such EBRs can be used for the consumption or production of different substances during circulation in the blood. The efficiency of EBRs may be affected by different factors, such as the permeability of the erythrocyte cell membrane to substrates and/or products of encapsulated enzymes, the stability of encapsulated enzymes inside erythrocytes, etc. In addition, the interaction of encapsulated enzymes with erythrocyte metabolic systems may affect the function of both encapsulated enzymes and metabolic systems. Many different applications of EBRs have been described in the literature, including the removal of asparagine from blood for antitumor therapy [3,4], removal of excess adenosine to treat adenosine deaminase deficiency [5], transformation of thymidine to thymine to treat thymidine phosphorylase deficiency [6,7], removal of ammonium [8,9,10,11] and alcohol (ethanol) [12,13,14,15,16,17] from blood in cases of intoxication, etc. 

The usefulness of mathematical models for the analysis and simulation of experimental data obtained with EBRs, as well as for analysis of the interaction between encapsulated enzymes and erythrocyte metabolic systems, was demonstrated in several our publications [10,17,18]. For instance, it has been shown previously that when encapsulated in erythrocytes, ammonium-neutralizing enzymes may significantly affect energy metabolism in cells [18]. In this work, for the first time, the effect of encapsulated enzymes on the metabolism of EBRs has been studied using mathematical modeling. It has been shown that the oxidation of NADPH in the glutamate dehydrogenase reaction causes an elevation of NAD levels that, in turn, activate metabolic flux through the pentosephosphate pathway. A significant activation of the pentosephosphate pathway in human erythrocytes can cause a decrease in [ADP] and, as a result, a decrease in the pyruvate kinase reaction rate, the disappearance of the steady state in the system, the permanent accumulation of glycolysis intermediates, and, finally, the osmotic destruction of EBRs [18,19].

In this study, using mathematical modeling, we investigated the effects of the interaction of glycolysis with enzymes encapsulated in EBRs prepared for the removal of alcohol from the blood. 

Excessive alcohol consumption is an important problem in modern society. Therefore, the fight against alcohol intoxication is not only an important social task, but also an urgent task in respect of modern medicine [20]. Several attempts have been described in the literature to produce EBRs that remove ethanol from the bloodstream [12,13,14,15,16,17]. The most effective and promising version of such EBRs involves those prepared with the simultaneous encapsulation of alcohol dehydrogenase (ADH) and acetaldehyde dehydrogenase (ALDH) [15,16,17]. In such EBRs, ethanol is first oxidized to acetaldehyde in an ADH reaction:ETH + NAD → ACALD + NADH,
where ETH denotes ethanol and ACALD denotes acetaldehyde. Next, the acetaldehyde produced in the ADH reaction is further oxidized to acetate in an ALDH reaction, which is irreversible under intracellular conditions [21]:ACALD + NAD → ACT + NADH.

Since glycolysis and the EBR-encapsulated ADH and ALDH enzymes use NAD and NADH as common substrates, they can affect each other via these metabolites. The general scheme demonstrating the interaction of glycolysis with the ADH and ALDH reactions via NAD and NADH (NAD/NADH loop) is shown in Figure 1.

The aim of the present study was to determine the possible effects of the interaction between glycolysis and enzymes encapsulated into erythrocytes ADH and ALDH on glycolysis itself, and on the efficiency of ethanol consumption by such erythrocytes (EBRs). In this work, we constructed and analyzed a mathematical model of EBRs, which describes the metabolic system shown in Figure 1. Analysis of the model revealed that the interaction of glycolysis with EBR-encapsulated ADH and ALDH enzymes via the NAD/NADH loop can cause metabolic oscillations and the disappearance of the steady state in glycolysis. This, in turn, can decrease the efficiency of alcohol consumption and cause EBRs’ death.

## 2. Results and Discussion

### 2.1. Kinetics of Ethanol Consumption

In the absence of ethanol, or with zero activity of ADH and ALDH, the model has a single steady state, with variable values close to those obtained in other models and in intact erythrocytes [10,22,23] (Table 1).

The simulation of experimental conditions in the presence of ethanol shows that EBRs can consume ethanol, either added to the blood or to the erythrocyte suspension, at a rate proportional to the activity of ADH and ALDH in the EBRs and to the ethanol concentration (Figure 2A,B).

The NAD concentration decreases in proportion to ADH and ALDH activity and ethanol concentration (Figure 2C). If EBRs contain only ADH, at the initial ethanol concentration of 10 mM, equilibrium in the ADH reaction is achieved after the consumption of 40 µM of ethanol and accumulation of 40 µM of acetaldehyde, and no further ethanol consumption can be achieved (Figure 2D). If we set the acetaldehyde concentration in the model equal to zero, the initial rate of ethanol consumption is slightly higher compared with the model in which the acetaldehyde concentration is calculated as an intermediate between the ADH and ALDH reactions (Figure 2D). However, the overall kinetics of ethanol consumption are almost the same in these two versions of the model (Figure 2D). Thus, our results reveal that EBRs loaded with ADH alone cannot provide efficient ethanol removal. For the efficient consumption of ethanol, EBRs must contain ALDH in addition to ADH. The model predicts very low acetaldehyde concentrations, which even decrease during ethanol consumption in EBRs containing both ADH and ALDH (Figure 2E). These low acetaldehyde levels decrease the reverse ADH reaction rate and thus provide efficient ethanol consumption in the model. The model successfully describes the results obtained earlier in in vitro experiments with EBRs containing ADH and ALDH [15] (Figure 2F).

### 2.2. Effects of ADH and ALDH Activity

Figure 3 shows the dependence of the steady-state ethanol consumption rate in EBRs on the activity of encapsulated ADH and ALDH (Figure 3A) and on the concentration of ethanol (Figure 3B). The graphs were created under the assumption that the ethanol concentration is constant. 

Analysis of the model shows that the steady-state rate of ethanol consumption increases with the increase in the activity of ADH and ALDH inside the EBRs, or with the increase in the ethanol concentration (Figure 3). At an ethanol concentration of 10 mM, the model has a single non-zero stable steady state in the range of ADH and ALDH activity from 0 to 101.5 mM/h (Figure 3A). The glycolysis rate remains constant, as well as most of the model variables, except for the concentrations of FDP, DAP, G3P, PYR, LAC, NAD, NADH, and acetaldehyde (Figure 4). The steady-state rates of the HK and PFK reactions are 1.12 mM/h, the steady-state rates of the GAPGH and PK reactions are 2.24 mM/h, and the steady-state rate of the PGK reaction is 1.61 mM/h. The rate of the PGK reaction is lower compared with the rates of the GAPDH and PK reactions, since part of the glycolytic flux bypasses the PGK via the 2,3-DPG shunt (Figure 1). After the increase in the ADH and ALDH activity above 101.5 mM/h, the non-zero steady state in the model becomes unstable and the system switches to an oscillation mode (Figure 4 and Figure 5). In Figure 5A–C, one can see synchronous oscillations in groups of metabolites interconnected via rapid, near-equilibrium enzymatic reactions. The amplitude and period of oscillations increase with the increase in the ADH and ALDH activity from 101.5 to 108.5 mM/h (Figure 6). Interestingly, the oscillations in the ethanol consumption rate (V_ADH_) are relatively small compared with the steady-state rate of ethanol consumption (Figure 5F). Thus, the steady-state rate can be used to characterize the rate of ethanol consumption in the oscillation mode. The increase in the ADH and ALDH activity above 108.5 mM/h causes a switch from the oscillation mode to an infinite accumulation of glycolysis intermediates, such as FDP, DAP, and GAP (Figure 7A). 

Finally, the steady state in the model disappears when the activity of the encapsulated enzymes is equal to 157 mM/h, which is also associated with the accumulation of glycolysis intermediates (Figure 7A).

### 2.3. Effects of Ethanol Concentration

Similar model behavior was observed with increasing ethanol concentration (Figure 3B). For ADH and ALDH activity of 40 mM/h, the model has one stable steady state in the ethanol concentration range from 0 to 31 mM. The increase in the ethanol concentration to above 31 mM causes the instability of the steady state, which switches the model into the oscillation mode. With a further increase in the ethanol concentration (above 34 mM), the oscillation mode is replaced by the infinite accumulation of glycolysis intermediates (FDP, DAP, and GAP). Eventually, the model loses its stationary state at an ethanol concentration of 59 mM.

### 2.4. Mechanism of Oscillations

The interaction of ethanol-consuming enzymes with glycolysis in the EBRs is provided via NAD, a common substrate for GAPDH, ADH, and ALDH. The increase in the ethanol consumption rate due to the increase in the ADH and ALDH activity, or due to the increase in ethanol concentration, causes a decrease in the NAD concentration (Figure 4E), and thus a temporal decrease in the GAPDH reaction rate (and the ATP production rate). This causes the accumulation of GAP, the second substrate of GAPDH (Figure 4B). 

The increased GAP concentration allows the rate of the GAPDH reaction (and the rate of ATP production) to be maintained at a normal physiological steady-state level, despite the decrease in [NAD]. Thus, an increase in the ethanol consumption rate in EBRs is associated with a decrease in [NAD] and an increase in [GAP] (Figure 3 and Figure 4).

The concentrations of GAP, DAP, and FDP are in equilibrium. These concentrations increase significantly (up to 0.26, 0.55, and 2.3 mM, respectively) with the increase in the ethanol consumption rate to 1.49 mM/h ([NAD] decreases to 0.031 mM), and small fluctuations in the concentration of NAD cause significant changes in the concentrations of GAP, DAP, and FDP (Figure 4B,E). As mentioned above, a decrease in [NAD] causes an accumulation of GAP that is associated with a decrease in the ATP production rate and, thus, with a decrease in ATP levels. An increase in [NAD] should cause an increase in GAP consumption in the GAPDH reaction and, thus, an increase in the ATP production rate and ATP levels. In this way, changes in [NAD] should cause changes in ATP levels as well as in the levels of ADP and AMP. Changes in ATP, ADP, and AMP concentrations should, in turn, affect the glycolytic flux at the levels of HK and PFK. Because of the high summary concentration of GAP, DAP, and FDP at the border of the oscillation mode, their concentrations change relatively slowly. Indeed, the normal glycolysis rate at the level of the PK reaction is about 2 mM/h. Therefore, it would take about one hour to increase the summary concentration of GAP, DAP, and FDP by 2 mM, even if the total glycolytic flux goes into these metabolites. Under these conditions, glycolysis cannot respond fast enough to changes in the system, which causes a loss of the steady-state stability and changes the system to an oscillation mode.

An analysis of the phase shifts between oscillations in the concentrations of different metabolites (Figure 8) allows us to suggest the following explanation for the oscillations. At high ATP concentrations, corresponding to low AMP concentrations in cells (Figure 8A), the PFK reaction is suppressed, since ATP is a strong inhibitor while AMP is a strong activator of this enzyme [24]. Under these conditions, the doubled rate of the PFK reaction is lower than the rate of the GAPDH reaction (Figure 8B), which causes a decrease in [GAP], a decrease in the GAPDH reaction rate, and a decrease in ATP production in the following glycolytic reactions. As a result, the ATP level decreases while the AMP level increases, which activates the PFK reaction and, finally, the doubled PFK reaction rate begins to exceed the GAPDH reaction rate. At this moment, the GAP level begins to increase, followed by an increase in the GAPDH reaction rate. This causes an increase in the ATP production rate, an increase in ATP concentration, a decrease in AMP concentration, and a decrease in the inhibition of the PFK reaction. Then, the next cycle begins. Thus, oscillations appear due to the interaction between the upper and lower parts of glycolysis under conditions where the decreased NAD level limits the GAPDH reaction rate and the ATP production rate in the lower part of glycolysis. Ethanol-consuming ADH and ALDH enzymes decrease the NAD levels. This decreases the GAPDH reaction rate, providing conditions for the oscillations in glycolysis.

Interestingly, within the oscillation mode, the model predicts the existence of a sharp transition between relatively fast low-amplitude oscillations and relatively slow high-amplitude oscillations (Figure 6). 

### 2.5. Accumulation of Glycolysis Intermediates

Above the ethanol consumption rate of 1.50 mM/h (with [NAD] decreased to 0.030 mM), the oscillation mode is replaced by an infinite accumulation of glycolysis intermediates. Finally, when the ethanol consumption rate increases to 1.55 mM/h (with [NAD] decreased to 0.025 mM), the increase in the GAP concentration cannot compensate for the decrease in the GAPDH reaction rate caused by the decrease in the NAD level. The steady state in the model disappears.

Interestingly, the rate of glycolytic intermediate accumulation decreases with the increasing activity of ADH and ALDH (Figure 7A). This happens because the accumulation of the glycolytic intermediates is associated with a significant decrease in ATP concentration (Figure 7B). Under these conditions, the PFK reaction rate decreases because [ATP] drops so significantly that PFK is limited by ATP as a substrate.

### 2.6. Conclusions

In conclusion, we note the following:

Along with ADH, ALDH is an essential component of EBRs, and is necessary to provide an efficient removal of ethanol from the blood. 

Our data show that the maximal rate of ethanol consumption by EBRs loaded with ADH and ALDH is about 1.5 mM/h (Figure 3). Attempts to increase this rate by increasing the activity of encapsulated ADH and ALDH in EBRs make no sense. Indeed, at very high ALD and ALDH activities, EBR glycolysis can move into the oscillation mode or into the endless accumulation of glycolytic intermediates. In the oscillation mode, the concentrations of metabolites can increase by more than 30 mM above their normal physiological levels (Figure 5 and Figure 6), which can cause the osmotic damage of EBRs [25]. Of course, the endless accumulation of glycolysis intermediates (FDP, DAP, and GAP) (Figure 7) will also cause such osmotic damage. We note that the oscillation mode, as well as the permanent accumulation of glycolysis intermediates, were obtained in the model under an assumption regarding the constant ethanol concentration, which is not the case in most real situations. However, these modes may be realistic in other EBR applications, when the function of the encapsulated enzymes is associated with the consumption of intracellular NAD.

A significant number of papers have been devoted to the theoretical and experimental study of oscillations in glycolysis [26,27,28,29,30]. However, the possibility of oscillations in human erythrocyte glycolysis, caused by the modulation of the GAPDH reaction rate by [NAD], has not been reported so far to our knowledge.

## 3. Mathematical Model

The mathematical model comprises a system of 16 ordinary differential equations and 3 algebraic equations describing the kinetics of concentrations of glycolysis intermediates, ATP, ADP, AMP, NAD, NADH, ethanol, and acetaldehyde (Table 2). The equations for the reaction rates in glycolysis and for the rates of ATP consumption, as well as the parameter values, were taken from [22] with a few modifications (Table 3). The model was supplemented with equations describing the pyruvate and lactate transport across the erythrocyte cell membrane (Table 2), in accordance with the information presented in [31]. Equations for the ADH and ALDH reaction rates and the parameter values were taken from [17,32,33]. All model equations describing the enzymatic reaction rates and the pyruvate and lactate transport across the cell membrane are presented in Table 3.

The following modifications in the glycolysis reaction rate equations were made:

TPI (α_TPI_), PGK (α_PGK_), and ENO (α_ENO_) activities were changed from 3000, 7330, and 83 mM/h [22] to more correct values of 19522, 2115, and 120 mM/h, respectively [34].

The following assumptions were made regarding the model:The rate of the hexokinase reaction does not depend on glucose concentration because the normal physiological glucose concentration in the blood is significantly larger than the value of the hexokinase Michaelis constant for glucose [35].The concentration of orthophosphate is constant and equal to 1.0 mM [22].The sums of concentrations [NAD] + [NADH] and [ATP] + [ADP] + [AMP] are constant and equal to 0.05 and 1.744 mM, respectively [22,36].The adenylate kinase reaction is in equilibrium because of the high activity of adenylate kinase in erythrocytes [22,36].The erythrocyte cell membrane is impermeable to glycolysis metabolites, except for glucose, pyruvate, and lactate [31,37].The extracellular concentrations of pyruvate ([PYR]ext) and lactate ([LAC]ext) are constant and equal to 0.07 and 1.2 mM, respectively [22].The permeability of the erythrocyte cell membrane for ethanol and acetate is high, and intracellular concentrations of these metabolites are equal to the extracellular concentrations [38,39].The extracellular concentration of acetate is zero.Acetaldehyde produced in the EBRs from ethanol does not leave the cells.The possible effect of the transmembrane potential on the transport of metabolites was not taken into account.The erythrocyte cell volume is constant.

All metabolite concentrations, enzyme activities, and enzymatic reaction rates were normalized per liter of cells (erythrocytes) and expressed in mM or mM/h.

All numeric solutions of the model system of equations were obtained in MATLAB (version 9.9.0 (R2020b)) using a variable-step variable-order solver based on numerical differentiation formulas from order 1 to 5 (the ode15s function) [40]. The MATLAB code of the model and files for in silico experiments are presented in a Appendix A. Steady-state values of concentrations and reaction rates were obtained as solutions of algebraic equation systems with a trust-region algorithm (fsolve function).

The kinetics of ethanol consumption by EBRs was simulated using the metabolite concentrations shown in Table 1 as initial values. The initial ACALD concentration was set equal to zero. Volume fractions occupied by erythrocytes and external medium were taken into account in such simulations.

## Figures and Tables

**Figure 1 ijms-24-10124-f001:**
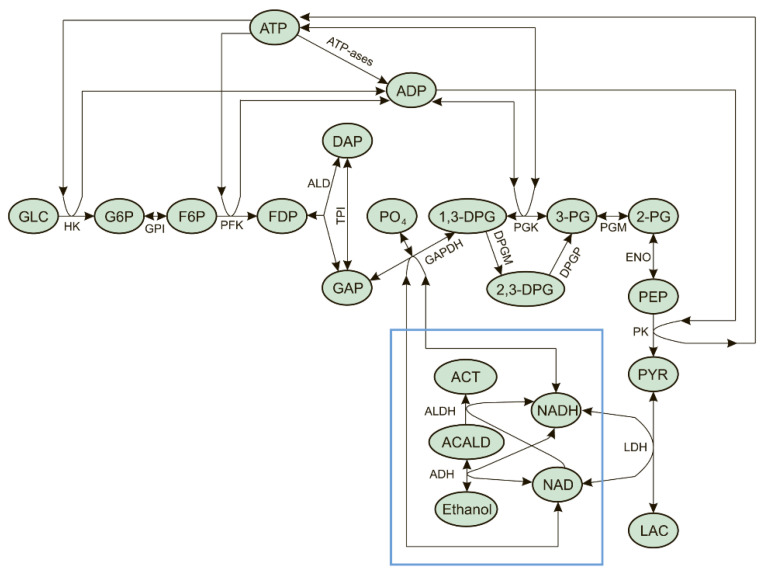
Interaction of glycolysis with encapsulated enzymes in erythrocyte-bioreactors consuming ethanol. The encapsulated enzymes are shown inside the blue frame. The following abbreviations are used: GLC—glucose, G6P—glucose-6-phosphate, F6P—fructose-6-phosphate, FDP—fructose-1,6-diphosphate, DAP—dihydroxyacetone phosphate, GAP—glyceraldehyde phosphate, 1,3-DPG—1,3-diphosphoglycerate, 2,3-DPG—2,3-diphosphoglycerate, 3-PG—3-phosphoglycerate, 2-PG—2-phosphoglycerate, PEP—phosphoenolpyruvate, PYR—pyruvate, LAC—lactate, PO_4_—orthophosphate, ACALD—acetaldehyde, ACT—acetate, HK—hexokinase, GPI—glucose-6-phosphate isomerase, PFK—phosphofructokinase, ALD—aldolase, TPI—triosephosphate isomerase, GAPDH—glyceraldehyde phosphate dehydrogenase, PGK—phosphoglycerate kinase, PGM—phosphoglycerate mutase, ENO—enolase, PK—pyruvate kinase, LDH—lactate dehydrogenase, ADH—alcohol dehydrogenase, and ALDH—acetaldehyde dehydrogenase. ATP-ases denotes the summary rate of ATP consumption by cell ATP-ases.

**Figure 2 ijms-24-10124-f002:**
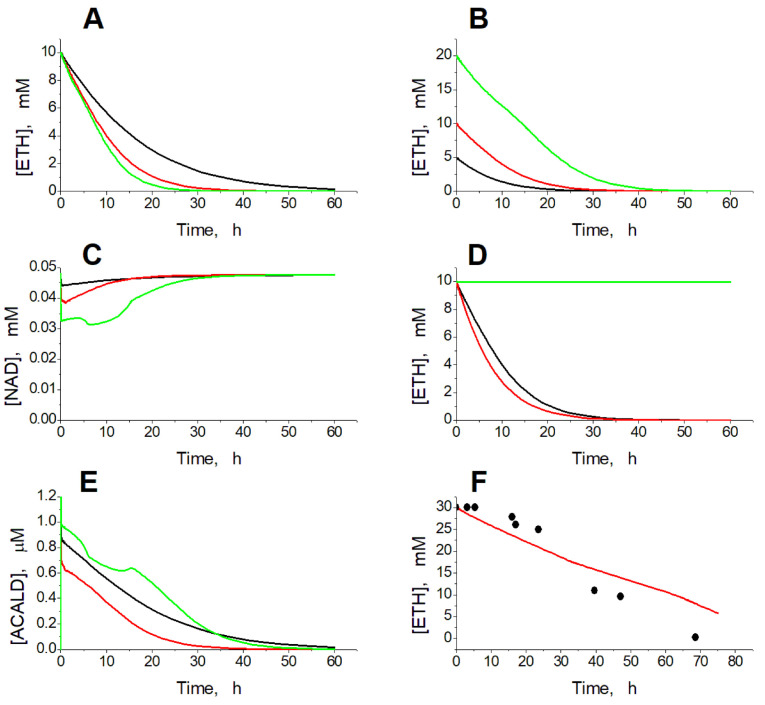
Kinetics of ethanol consumption in the model. (**A**) Simulated dependence of ethanol concentration on time in a 50% EBR suspension at ADH and ALDH activities equal to 30 (black line), 60 (red line), and 90 (green line) mM/h after addition of ethanol to a final concentration of 10 mM. (**B**) Simulated dependence of ethanol concentration on time in the 50% EBR suspension at ADH and ALDH activities equal to 60 mM/h after addition of ethanol to a final concentration of 5 mM (black line), 10 mM (red line), and 20 mM (green line). (**C**) Kinetics of NAD concentration in the model at activity of the encapsulated enzymes equal to 30 mM/h with 10 mM ethanol (black line), 60 mM/h with 10 mM ethanol (red line), and 60 mM/h with 20 mM ethanol (green line). (**D**) Simulated dependence of ethanol concentration on time in the EBR suspension (hematocrit 50%) at ADH activity equal to 60 mM/h. Activity of ALDH was equal to 60 mM/h (black and red lines) or to zero (green line). Acetaldehyde concentration was calculated from the model (black and green lines) or set to zero (red line). (**E**) Kinetics of acetaldehyde concentration in the model under conditions described for panel C. (**F**) Experimental dependence of ethanol concentration on time obtained in [15] for the 25% suspension of human EBRs loaded with ADH (78 mM/h) and ALDH (18 mM/h), after addition of ethanol to a final concentration of 30 mM (circles), and simulation results obtained for the same parameter values (line).

**Figure 3 ijms-24-10124-f003:**
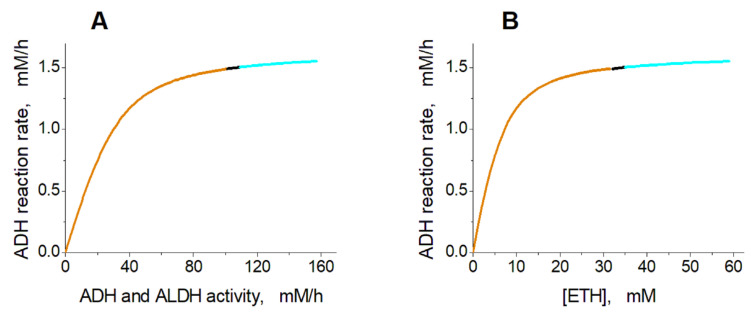
Dependence of the steady-state ethanol consumption rate in the model on the activity of ADH and ALDH and on the ethanol concentration. (**A**) The ethanol consumption rate was calculated at an ethanol concentration of 10 mM, assuming that the ADH activity was equal to the ALDH activity. (**B**) The ethanol consumption rate was calculated assuming that the ethanol concentration was constant and the activities of ADH and ALDH were equal to 40 mM/h. The orange, black, and cyan lines indicate stable steady states, unstable steady states with oscillation mode, and unstable steady states with metabolite accumulation, respectively.

**Figure 4 ijms-24-10124-f004:**
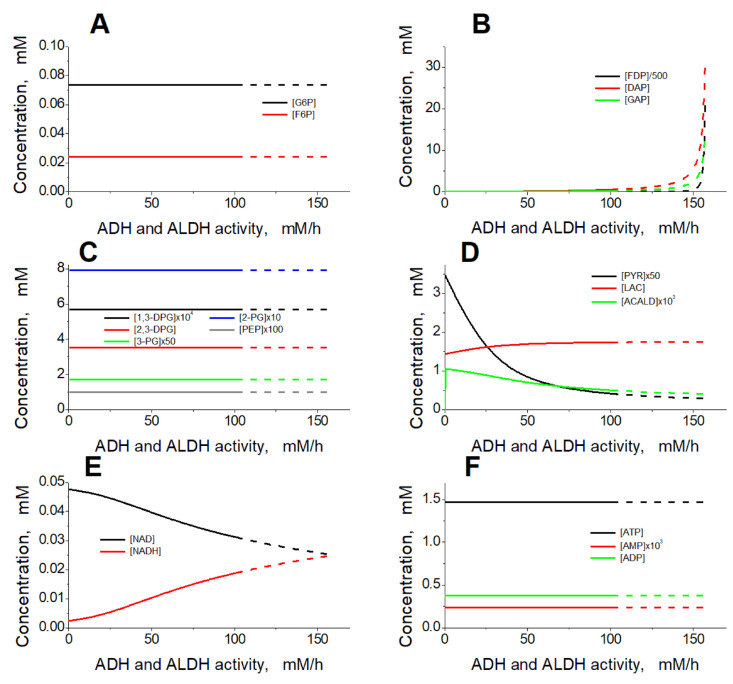
Dependence of the model’s steady state on the activity of ADH and ALDH. The following concentrations are shown (in mM): (**A**) G6P and F6P; (**B**) FDP/500, DAP and GAP; (**C**) 1,3-DPG × 10^4^, 2,3-DPG, 3-PG × 50, 2-PG × 10 and PEP × 100; (**D**) PYR × 50, LAC and ACALD × 10^3^; (**E**) NAD and NADH; and (**F**) ATP, AMP × 10^3^ and ADP. The ethanol concentration was constant and equal to 10 mM. The activity of ADH was equal to the ALDH activity. Dashed lines correspond to unstable steady states.

**Figure 5 ijms-24-10124-f005:**
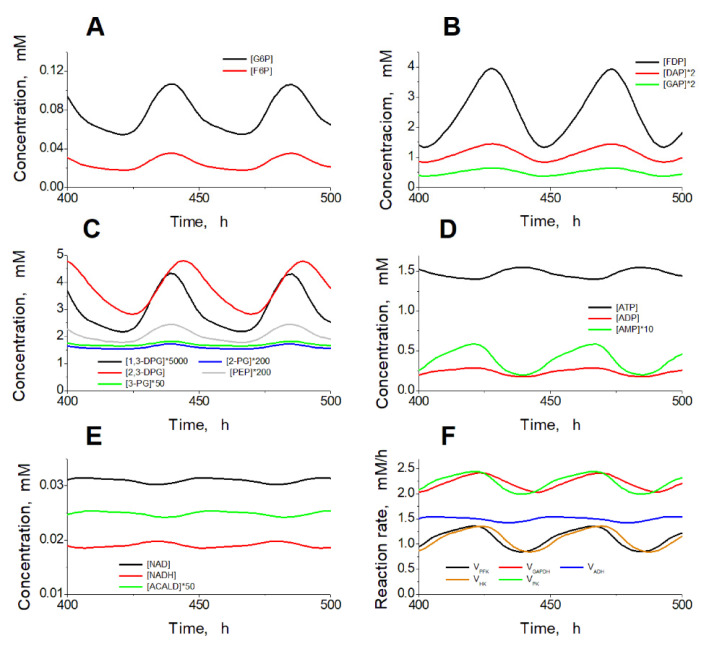
Steady oscillations of the metabolite concentrations and enzymatic reaction rates in the model. Oscillations of the metabolite concentrations in the upper part of glycolysis (**A**), the lower part of glycolysis (**B**,**C**), adenine nucleotides (**D**), NAD, NADH, and acetaldehyde (**E**), and enzymatic reaction rates (**F**). Calculations were performed at the constant ethanol concentration of 10 mM and with ADH and ALDH activities equal to 102.5 mM/h.

**Figure 6 ijms-24-10124-f006:**
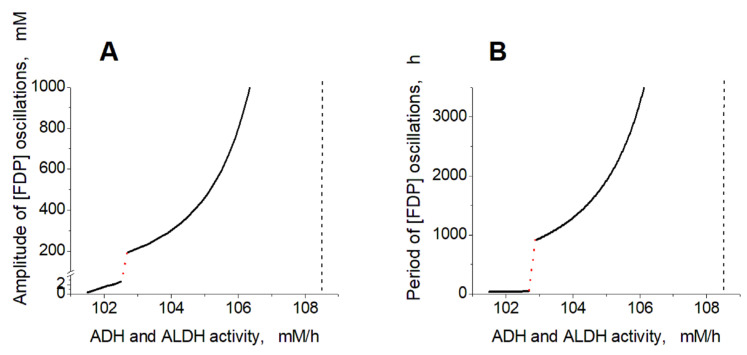
Dependence of oscillation parameters in the model on the ADH and ALDH activity. The dependence of [FDP] oscillation amplitude (**A**) and period (**B**) on the ADH and ALDH activity. The red dotted lines show a switch of the model between fast low-amplitude and slow high-amplitude oscillations. Vertical black dashed lines indicate a border of the oscillation mode at high activity of the encapsulated enzymes. The ethanol concentration was constant and equal to 10 mM. The activity of ADH was equal to the ALDH activity.

**Figure 7 ijms-24-10124-f007:**
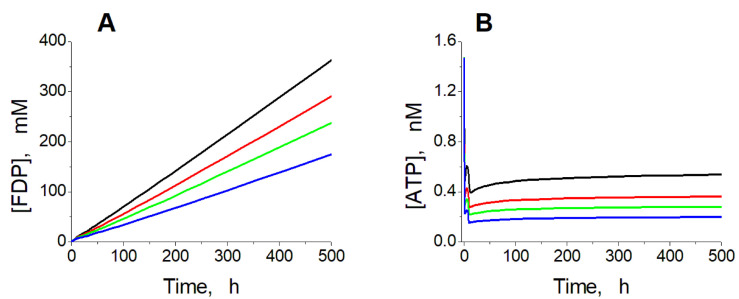
Kinetics of [FDP] (**A**) and [ATP] (**B**) in the model at high ADH and ALDH activity. Calculations were performed at the constant ethanol concentration of 10 mM, and ADH and ALDH activity equal to 120 (black line), 150 (red line), 180 (green line), and 240 (blue line) mM/h.

**Figure 8 ijms-24-10124-f008:**
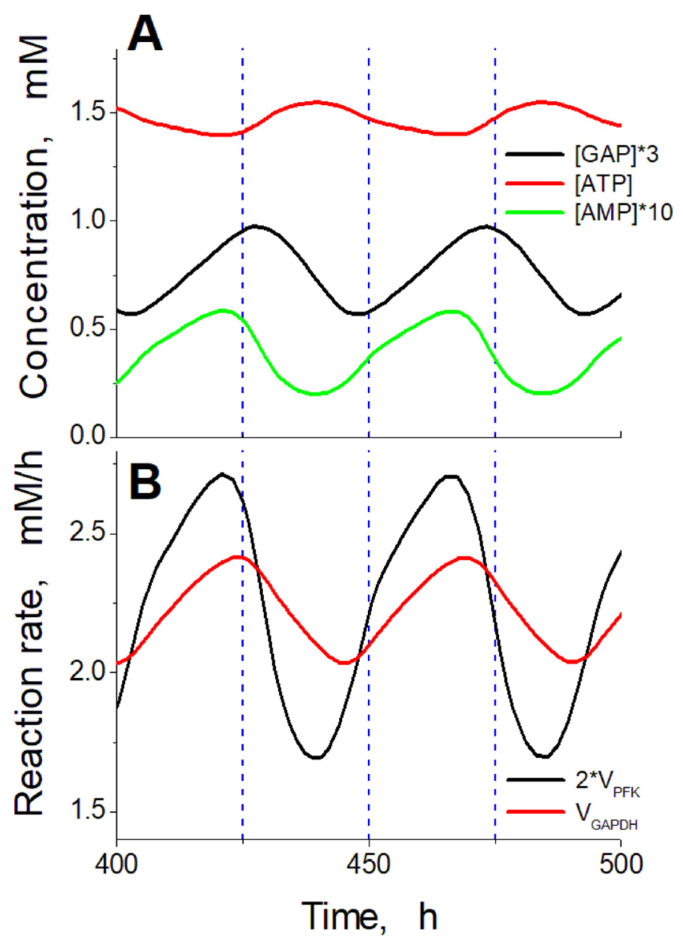
Time shifts between principal metabolite concentrations and reaction rates during steady oscillations in the model. (**A**) Comparison of ATP concentration with GAP concentration multiplied by 3 and AMP concentration multiplied by 10. (**B**) Comparison of doubled PFK reaction rate and GAPDH reaction rate. Calculations were carried out at a constant ethanol concentration of 10 mM and under ADH and ALDH activities of 102.5 mM/h.

**Table 1 ijms-24-10124-t001:** The steady-state concentrations of metabolites obtained in the model in the absence of ethanol and/or at zero activity of ADH and ALDH.

Metabolite ^(a)^	Concentration, mM	Metabolite ^(a)^	Concentration, mM
G6P	0.073	PEP	0.01
F6P	0.024	PYR	0.07
FDP	0.0084	LAC	1.43
DAP	0.034	NAD	0.048
GAP	0.015	NADH	0.002
1,3-DPG	5.7 × 10^−4^	ATP	1.471
2,3-DPG	3.5	ADP	0.235
3-PG	0.034	AMP	0.038
2-PG	0.008	ACALD	0

^(a)^ Abbreviations are described in the legend to Figure 1.

**Table 2 ijms-24-10124-t002:** Differential and algebraic equations describing the kinetics of metabolite concentrations in the model of erythrocyte-bioreactors prepared for ethanol consumption.

№	Variable ^(a)^	Equation ^(b)^
1	[G6P]	$\frac{d[G6P]}{dt}=V_{HK}-V_{PGI}$
2	[F6P]	$\frac{d[F6P]}{dt}=V_{PGI}-V_{PFK}$
3	[FDP]	$\frac{d[FDP]}{dt}=V_{PFK}-V_{ALD}$
4	[DAP]	$\frac{d[DAP]}{dt}=V_{ALD}-V_{TPI}$
5	[GAP]	$\frac{d[GAP]}{dt}=V_{ALD}+V_{TPI}-V_{GAPDH}$
6	[1,3-DPG]	$\frac{d[1,3-DPG]}{dt}=V_{GAPDH}-V_{DPGM}-V_{PGK}$
7	[2,3-DPG]	$\frac{d[2,3-DPG]}{dt}=V_{DPGM}-V_{DPGP}$
8	[3-PG]	$\frac{d[3-PG]}{dt}=V_{PGK}+V_{DPGD}-V_{PGM}$
9	[2-PG]	$\frac{d[2-PG]}{dt}=V_{PGM}-V_{ENO}$
10	[PEP]	$\frac{d[PEP]}{dt}=V_{ENO}-V_{PK}$
11	[PYR]	$\frac{d[PYR]}{dt}=V_{PK}-V_{LDH}+V_{trPYR}$
12	[LAC]	$\frac{d[LAC]}{dt}=V_{LDH}+V_{trLAC}$
13	[NAD]	$\frac{d[NAD]}{dt}=V_{LDH}-V_{GAPDH}-V_{ADH}-V_{ALDH}$
14	Energy charge (*Φ*) ^(c)^	$\frac{d\Phi }{dt}=\frac{1}{2P}(V_{PGK}+V_{PK}-V_{HK}-V_{PFK}-V_{ATP}-V_{ATPNa})$
15	[ETH]	$\frac{d[ETH]}{dt}=-V_{ADH}$
16	[ACALD]	$\frac{d[ACALD]}{dt}=V_{ADH}-V_{ALDH}$
17	Adenylate pool	[*ATP*]+[*ADP*]+[*AMP*] = *P*
18	Adenylate kinase equilibrium	$\frac{\left[ATP\right][ADP]}{{\left[ADP\right]}^{2}}=1$
19	NAD/NADH pool	[*NAD*] + [*NADH*] = *N*

^(a)^ The abbreviations are described in the legend to Figure 1. ^(b)^ The right parts of the differential equations represent the sum of the rates of reactions, which produce and consume the corresponding metabolites, and are denoted as *V_X_*, where *_X_* is the abbreviation for the corresponding enzyme. The right part of the energy charge equation includes the rates of ATP consumption by transport Na^+^/K^+^-ATP-ase (*V_ATPNa_*) and additional ATP-ase (*V_ATP_*). The right parts of the equations for pyruvate and lactate also include the rates of pyruvate and lactate transport across the erythrocyte cell membrane (*V_trPYR_* and *V_trLAC_*, respectively). ^(c)^ Energy charge *Φ* = (2[*ATP*] + [*ADP*])/(2([*ATP*] + [*ADP*] + [*AMP*])).

**Table 3 ijms-24-10124-t003:** Model equations for the rates of enzymatic reactions.

Hexokinase [22] $V_{HK}=a_{HK}\frac{\left[ATP\right]/K_{HK}^{1}}{1+\frac{\left[ATP\right]}{K_{HK}^{1}}+\frac{\left[G6P\right]}{K_{HK}^{2}}}$ $a_{HK}=12$ mM/h, $K_{HK}^{1}=1$ mM, $K_{HK}^{1}=5.5\cdot 10^{-3}$ mM
Glucose-6-phosphate isomerase [22] $V_{GPI}=a_{GPI}\frac{\left(\left[G6P\right]-\left[F6P\right]K_{GPI}^{1}\right)/K_{GPI}^{2}}{1+\frac{\left[G6P\right]}{K_{GPI}^{2}}+\frac{\left[F6P\right]}{K_{GPI}^{3}}}$ $a_{GPI}=360$ mM/h, $K_{GPI}^{1}=3$ mM, $K_{GPI}^{2}=0.3$ mM, $K_{GPI}^{3}=0.2$ mM
Phosphofructokinase [22] $V_{PFK}=a_{PFK}\frac{1.1\cdot [ATP][F6P]\left(\frac{1}{1+\left[AMP\right]/K_{PFK}^{3}}+\frac{2[AMP]}{K_{PFK}^{3}+[AMP]}\right)}{\left(K_{PFK}^{2}+[ATP]\right)\left(K_{PFK}^{1}+[F6P]\right)\left(1+10^{8}\frac{{\left(1+\left[ATP\right]/K_{PFK}^{4}\right)}^{4}}{{\left(1+\left[AMP\right]/K_{PFK}^{3}\right)}^{4}{\left(1+\left[F6P\right]/K_{PFK}^{5}\right)}^{4}}\right)}$ $a_{PFK}=380$ mM/h, $K_{PFK}^{1}=0.1$ mM, $K_{PFK}^{2}=2$ mM, $K_{PFK}^{3}=0.01$ mM, $K_{PFK}^{4}=0.195$ mM, $K_{PFK}^{5}=3.7\cdot 10^{-4}$ mM
Aldolase [22] $V_{ALD}=a_{ALD}\frac{\frac{\left[FDP\right]}{K_{ALD}^{1}}-\frac{\left[DAP\right][GAP]}{K_{ALD}^{2}}}{1+\frac{\left[FDP\right]}{K_{ALD}^{3}}+\frac{\left[DAP\right]}{K_{ALD}^{4}}+\frac{\left[GAP\right]}{K_{ALD}^{5}}+\frac{\left[FDP\right][DAP]}{K_{ALD}^{3}K_{ALD}^{4}}+\frac{{\left[DAP\right]}^{2}}{K_{ALD}^{4}K_{ALD}^{6}}+\frac{\left[DAP\right][GAP]}{K_{ALD}^{4}K_{ALD}^{7}}}$ $a_{PFK}=76$ mM/h, $K_{ALD}^{1}=2\cdot 10^{-4}$ mM, $K_{ALD}^{2}=1.2\cdot 10^{-5}$ mM^2^, $K_{ALD}^{3}=0.01$ mM, $K_{ALD}^{4}=0.032$ mM, $K_{ALD}^{5}=2.1\cdot 10^{-3}$ mM, $K_{ALD}^{6}=2$ mM, $K_{ALD}^{7}=0.065$ mM
Triosephosphate isomerase [22,34] $V_{TPI}=a_{TPI}\frac{\left(\left[DAP\right]-[GAP]/K_{TPI}^{2}\right)/K_{TPI}^{1}}{1+\frac{\left[DAP\right]}{K_{TPI}^{1}}+\frac{\left[GAP\right]}{K_{TPI}^{3}}}$ $a_{TPI}=19,522$ mM/h, $K_{TPI}^{1}=0.82$ mM, $K_{TPI}^{2}=0.45$ mM, $K_{TPI}^{3}=0.43$ mM
Glyceraldehyde phosphate dehydrogenase [22] $V_{GAPDH}=a_{GAPDH}\frac{\left(\left[GAP\right]\left[NAD\right]\left[P_{i}\right]-[1,3DPG][NADH]/K_{GAPDH}^{4}\right)/K_{GAPDH}^{1}K_{GAPDH}^{2}K_{GAPDH}^{3}}{1.29\left(1+\frac{\left[GAP\right]}{K_{GAPDH}^{1}}+\frac{\left[1,3DPG\right]}{K_{GAPDH}^{5}}\right)\left(1+\frac{\left[NAD\right]}{K_{GAPDH}^{2}}+\frac{\left[NADH\right]}{K_{GAPDH}^{6}}\right)}$ $a_{GAPDH}=690$ mM/h, $K_{GAPDH}^{1}=0.13$ mM, $K_{GAPDH}^{2}=0.13$ mM, $K_{GAPDH}^{3}=3.4$ mM, $K_{GAPDH}^{4}=0.136$ mM, $K_{GAPDH}^{5}=0.013$ mM, $K_{GAPDH}^{6}=2\cdot 10^{-3}$ mM
Phosphoglycerate kinase [22,34] $V_{PGK}=a_{PGK}\frac{\left(\left[1,3-DPG\right]\left[ADP\right]-\frac{\left[3-PG\right]\left[ATP\right]}{K_{PGK}^{3}}\right)/(K_{PGK}^{1}K_{PGK}^{2})}{1+\frac{\left[ATP\right]}{K_{PGK}^{5}}+\frac{\left[ADP\right]}{K_{PGK}^{2}}+\frac{A\left[1,3-DPG\right]}{K_{PGK}^{1}}+\frac{B\left[3-PG\right]}{K_{PGK}^{6}}}$ $A=\left(K_{PGK}^{4}+\left[ADP\right]+K_{PGK}^{4}\left[ATP\right]/K_{PGK}^{5}\right)/K_{PGK}^{2}$ $B=\left(K_{PGK}^{7}+\left[ATP\right]+K_{PGK}^{7}\left[ADP\right]/K_{PGK}^{2}\right)/K_{PGK}^{5}$ $a_{PGK}=2115$ mM/h, $K_{PGK}^{1}=2.2\cdot 10^{-3}$ mM, $K_{PGK}^{2}=0.14$ mM, $K_{PGK}^{3}=380$ mM, $K_{PGK}^{4}=0.3$ mM, $K_{PGK}^{5}=0.27$ mM, $K_{PGK}^{6}=1.4$ mM, $K_{PGK}^{7}=0.4$ mM
Diphosphoglycerate mutase [22] $V_{DPGM}=a_{DPGM}\frac{\left[1,3-DPG\right]}{K_{DPGM}^{1}+K_{DPGM}^{2}\left[1,3-DPG\right]+[2,3-DPG]}$ $a_{DPGM}=3892$ mM/h, $K_{DPGM}^{1}=0.04$ mM, $K_{DPGM}^{2}=0.013$ mM
Diphosphoglycerate phosphatase [22] $V_{DPGP}=a_{DPGP}\frac{\left[2,3-DPG\right]}{\left[2,3-DPG\right]+K_{DPGD}^{1}\left(1+\left[2-PG\right]+\left[3-PG\right]/K_{DPGD}^{2}\right)}$ $a_{DPGP}=0.65$ mM/h, $K_{DPGP}^{1}=0.02$ mM, $K_{DPGM}^{1}=6\cdot 10^{-3}$ mM
Phosphoglycerate mutase [22] $V_{PGM}=a_{PGM}\frac{\left(\left[3-PG\right]-[2-PG]/K_{PGM}^{2}\right)/K_{PGM}^{1}}{1+\frac{\left[3-PG\right]}{K_{PGM}^{1}}+\frac{\left[2-PG\right]}{K_{PGM}^{3}}}$ $a_{PGM}=1100$ mM/h, $K_{PGM}^{1}=0.27$ mM, $K_{PGM}^{2}=0.24$ $K_{PGM}^{3}=0.02$ mM
Enolase [22,34] $V_{ENO}=a_{ENO}\frac{\left(\left[2-PG\right]-[PEP]/K_{ENO}^{2}\right){/K}_{ENO}^{1}}{1+\frac{\left[2-PG\right]}{K_{ENO}^{1}}+\frac{\left[PEP\right]}{K_{ENO}^{3}}}$ $a_{ENO}=120$ mM/h, $K_{ENO}^{1}=0.056$ mM, $K_{ENO}^{2}=6.7$, $K_{ENO}^{3}=2\cdot 10^{-3}$ mM
Pyruvate kinase [22] $V_{PK}=a_{PK}\frac{\left[PEP\right]\left[ADP\right]/(K_{PK}^{1}K_{PK}^{2})}{1+\frac{\left[ATP\right]}{K_{PK}^{3}}+\frac{\left[ADP\right]}{K_{PK}^{2}}+\frac{\left[PEP\right]}{K_{PK}^{1}}+\frac{[PEP]\left[ADP\right]}{K_{PK}^{1}K_{PK}^{2}}}$ $a_{ENO}=120$ mM/h, $K_{PK}^{1}=0.05$ mM, $K_{PK}^{2}=0.43$ mM, $K_{PK}^{3}=0.35$ mM
Lactate dehydrogenase [22] $V_{LDH}=a_{LDH}\frac{\left(\left[PYR\right]\left[NADH\right]-\frac{\left[LAC\right]\left[NAD\right]}{K_{LDH}^{3}}\right)/(K_{LDH}^{1}K_{LDH}^{2})}{1+\frac{\left[PYR\right]}{K_{LDH}^{1}}+\frac{K_{LDH}^{4}\left[NADH\right]}{K_{LDH}^{1}K_{LDH}^{2}}+\frac{[PYR]\left[NADH\right]}{K_{LDH}^{1}K_{LDH}^{2}}+\frac{K_{LDH}^{4}[LAC]\left[NADH\right]}{K_{LDH}^{1}K_{LDH}^{2}K_{LDH}^{5}}+\frac{C}{K_{LDH}^{5}K_{LDH}^{6}}}$ $C=K_{LDH}^{7}\left[NAD\right]+K_{LDH}^{6}\left[LAC\right]+\left[NAD\right]\left[LAC\right]+K_{LDH}^{7}\left[PYR\right]\left[NAD\right]/K_{LDH}^{1}$ $a_{LDH}=550$ mM/h, $K_{LDH}^{1}=0.022$ mM, $K_{LDH}^{2}=7\cdot 10^{-3}$ mM, $K_{LDH}^{3}=426$, $K_{LDH}^{4}=0.14$ mM, $K_{LDH}^{5}=380$ mM, $K_{LDH}^{6}=0.1$ mM, $K_{LDH}^{7}=170$ mM
Na^+^/K^+^-ATPase [22] $V_{NaKATP}=a_{NaKATP}\left[{Na}^{+}\right][ATP]$ $a_{NaKATP}=0.045$ mM/h, $\left[{Na}^{+}\right]=10\;mM$ [30]
All other ATPases are presented by the following equation [22] $V_{ATP}=a_{ATP}\frac{\left[ATP\right]}{\left[ATP\right]+K_{ATP}}$ $a_{NaKATP}=1.6$ mM/h, $K_{ATP}=1$ mM
Alcohol dehydrogenase [32,33] $V_{ADH}=\alpha_{ADH}\frac{\left[NAD\right]\left[ETH\right]-\frac{\left[ACALD\right][NADH]}{K_{ADH}^{eq}}}{D_{ADH}}$ $\begin{array}{ll} D_{ADH}=K_{ADH}^{2}K_{ADH}^{3} & +K_{ADH}^{3}\left[NAD\right]+K_{ADH}^{1}\left[ETH\right]+\left[NAD\right]\left[ETH\right]+\frac{K_{ADH}^{2}K_{ADH}^{3}}{K_{ADH}^{8}}\left[NADH\right]\\ & +\frac{K_{ADH}^{2}K_{ADH}^{3}K_{ADH}^{7}}{K_{ADH}^{5}K_{ADH}^{8}}\left[ACALD\right]+\frac{K_{ADH}^{2}K_{ADH}^{3}}{K_{ADH}^{5}K_{ADH}^{8}}\left[ACALD\right]\left[NADH\right]\\ & +\frac{K_{ADH}^{3}K_{ADH}^{7}}{K_{ADH}^{5}K_{ADH}^{8}}\left[ACALD\right]\left[NAD\right]+\frac{K_{ADH}^{1}}{K_{ADH}^{8}}\left[ETH\right]\left[NADH\right]\\ & +\frac{\left[ACALD\right]\left[ETH\right][NAD]}{K_{ADH}^{6}}+\frac{K_{ADH}^{2}K_{ADH}^{3}}{K_{ADH}^{4}K_{ADH}^{5}K_{ADH}^{8}}\left[ACALD\right]\left[ETH\right][NADH] \end{array}$ $K_{ADH}^{1}=0.074\;mM,K_{ADH}^{2}=0.61\;mM,K_{ADH}^{3}=13\;mM,K_{ADH}^{4}=0.43\;mM,$ $K_{ADH}^{5}=0.78\;mM,K_{ADH}^{6}=0.67\;mM,K_{ADH}^{7}=0.11\;mM,K_{ADH}^{8}=0.018\;mM,$ $K_{ADH}^{eq}=2\cdot 10^{-4},$ $\alpha_{ADH}—$ ADH activity—this is a varied parameter in the model.
Acetaldehyde dehydrogenase [21] $V_{ALDH}=\alpha_{ALDH}\frac{\left[NAD\right][ACALD]}{\left(\left[NAD\right]\left[ACALD\right]+K_{ALDH}^{3}\left[NAD\right]+K_{ALDH}^{1}\left[ACALD\right]+K_{ALDH}^{2}K_{ALDH}^{3}\right)\left(1+\frac{\left[NADH\right]}{K_{ALDH}^{4}}\right)}$ $K_{ALDH}^{1}=0.02\;mM,K_{ALDH}^{2}=0.07\;mM,K_{ALDH}^{3}=0.009\;mM,K_{ALDH}^{4}=0.1\;mM$ $\alpha_{ALDH}$—ALDH activity—this is a varied parameter in the model
Pyruvate transport [31] $V_{trPYR}=A_{trPYR}\frac{{\left[PYR\right]}_{ext}-[PYR]}{{\left[PYR\right]}_{ext}+\left[PYR\right]+\left(1+\frac{\left[LAC\right]+{\left[LAC\right]}_{ext}}{K_{I}^{trPYR}}\right)K_{m}^{trPYR}}$ $A_{trPYR}=120\;mM/h$; $K_{m}^{trPYR}=1.9\;mM$; $K_{I}^{trPYR}=11\;mM$
Lactate transport [31] $V_{trLAC}=A_{trLAC}\frac{{\left[LAC\right]}_{ext}-[LAC]}{{\left[LAC\right]}_{ext}+\left[LAC\right]+\left(1+\frac{\left[PYR\right]+{\left[PYR\right]}_{ext}}{K_{I}^{trLAC}}\right)K_{m}^{trLAC}}$ $A_{trLAC}=120\;mM/h$; $K_{m}^{trLAC}=9\;mM$; $K_{I}^{trLAC}=1.6\;mM$

## Data Availability

All data generated or analyzed during this study are included in the published article.

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
