# Peer review of "Prediction of Oscillations in Glycolysis in Ethanol-Consuming Erythrocyte-Bioreactors"

_ijms, 2023, doi:10.3390/ijms241210124_

Round 1
Reviewer 1 Report
The manuscript presents an ODE model to describe the metabolism changes when introducing two encapsulated enzymes into erythrocytes for ethanol conversion. Although the topic has potential interest to the readership of the journal, the modeling approach is poorly described and the model output is poorly presented. Here are the specific points to improve:
1. The results and discussion section should start with a brief description of metabolic pathway, modeling approach, major assumptions and experimental conditions they attempt to simulate and why. Starting directly with model output decreases the readability of the paper.
2. The lack of segmention, long sentences, complex wording and overwhelming abbreviations and figures make the results section very hard to read. I suggest deviding it into subsection, for instance: "baseline model", "combining ADH and ALDH lead to ethanol removel", "steady state under low ADH/ALDH/ethanol", "switching to oscillation mode", "possible explanation for osciallatory state"... Authors should reduce the number of figures and better organize them in the text to improve readability.
3. Line 171-178, authors claim that "oscillatory behavior was observed also with increasing ethanol concentration". However there's no results or figures linked to this claim.
4. For the explanation of oscillatory state, at paragraph 205-209, authors mention about the changes of GAP/DAP/FDP due to decreased NAD, then in the paragraph 216-233, they seem to give another explanation related to ATP/AMP, and the paragraph is overwhelmed with increase/decrease, thus very hard to follow. Please reformulate the two paragraphs.
5. The simulation results of the entire manuscript is hardly supported by any in vitro experimental data, making the modelling approach less convincing.
6. This is a typical unreproducible work. Please consider sharing the Matlab code of the model, along with all in silico experiments they performed.
Please consider using shorter sentences. Please reduce redundant wording such as "decrease", "increase" throughout the manuscript.
Author Response
Please sea the attachment

Reviewer 2 Report
The manuscript entitled “Prediction of oscillations in glycolysis in ethanol consuming erythrocyte-bioreactors” presented mathematical models of energy metabolism in erythrocyte-bioreactors loaded with alcohol dehydrogenase and acetaldehyde dehydrogenase. The manuscript is primarily theoretical and lacks experimental data (in vitro). The current form of manuscript does not meet the Journal criteria. As a result, it is recommended for Major revision.
The specific comments, which could help to improve the manuscript are:
1. The manuscript should be revised for grammatical & punctuation errors.
2. In Introduction section: Authors should discuss their earlier related findings and research envisaged.
3. The presentation of method section should be improved.
4. The study lacks any in-lab practical work (in vitro/enzyme assay). In vitro assay should be performed to validate the results of theoretical predictions.
5. Figure 1 seems similar to the earlier publication of the author. Please refer https://doi.org/10.3390/metabo11010036
6. The conclusion part should be improved to show the highlights of research.
Minor editing of English language required
Reviewer 3 Report
Dear authors, yours contributions are interessting for society and medical science.
Unfortunately, erythrocyte membrane injuries were soon evaluated in different addictions (alcohol including), leading to hemorhelogical and hematometrical changes in RBCs, related to bad outcome of (alcohol=) pathological state (E Zvetkova, D Fuchs, 2017).That means: Modelling presented in the paper is near to the real life situation and could be useful in clinical practice. Other questions - related to the Model cteated by specialists, could be re-directed to the authots.
Congratulations!
Youts sincerely, Reviewer
Round 2
Reviewer 2 Report
The overall concept of the manuscript is good. The authors have justified most of the comments. However, comment number 4 and 5 still need to be justified.
Comment 4. The manuscript corresponds to the subject of the special issue of the journal “Development in Drug Discovery: Computational and Experimental Aspects”. As per understanding this issue includes experimental Aspects which refers to lab experiments (in vitro/in vivo). Authors are suggested to refer their own article for the in-vitro experiments. https://doi.org/10.1038/s41598-018-37828-5
Comment 5. Authors are suggested to change the way of presentation for Figure 1 as a part of it seems similar to the earlier publication of the authors. Please refer https://doi.org/10.3390/metabo11010036; https://doi.org/10.1038/s41598-018-37828-5
